# Generalizable Transferability Estimation of Foundation Vision Models via Implicit Learning

## Abstract

Transferability estimation aims to identify the most suitable model from a collection of pre-trained models for specific downstream tasks, playing a crucial role in the success of the pre-training and fine-tuning paradigm. However, the recent proliferation of pre-trained models with diverse architectures and training strategies poses significant challenges for transferability estimation due to discrepancies in intrinsic model characteristics, making it difficult for existing methods to accurately simulate embedding space evolution within feasible computational limits. To address these challenges, we propose an Implicit Transferability Modeling (ITM) paradigm that incorporates an implicit modeling strategy for the intrinsic properties of pre-trained models, enabling more accurate transferability estimation. ITM employs a Divide-and-Conquer Adaptation (DCA) process to efficiently model the transfer process, reducing both learning complexity and computational cost. Additionally, we introduce a Pseudo-Clustering-based Optimization (PCO) strategy combined with static and dynamic constraints, enabling effective estimation without intensive retraining. Our method significantly outperforms state-of-the-art approaches, achieving notable improvements across ten widely used benchmarks and demonstrating its effectiveness and generalizability in enabling accurate and efficient model selection for downstream tasks.

## 1 Introduction

Recently, the pre-training and fine-tuning paradigm has achieved remarkable success in numerous computer vision applications, making Transferability Estimation (TE) a significant topic that involves predicting the performance of various pre-trained models on downstream tasks within a limited time frame.

Early methods (Tran et al., 2019; Nguyen et al., 2020) modeled the joint distribution between pre-trained labels and downstream task labels. In contrast, methods like LogME (You et al., 2021) and ETran (Gholami et al., 2023) evaluate models using embedding space and downstream task labels, extending transferability estimation to self-supervised approaches. Subsequently, Li et al. (2023) and Hu et al. (2024) recognized that these static methods overlook the dynamic changes in models during the fine-tuning process. They proposed modeling the dynamic aspects of fine-tuning to map the embedding spaces before and after the process, leading to improved performance.

However, with the rise of diverse self-supervised pre-training strategies and increasingly sophisticated network architectures, existing methods encounter discrepancies from various model collections, leading to inaccurate predictions and unreliable model selection for downstream tasks, as illustrated in Fig. 1. Recent models pre-trained with techniques such as Instance Discrimination (ID) (Chen et al., 2021; Caron et al., 2021) and Masked Image Modeling (MIM) (He et al., 2022; Xie et al., 2022) exhibit divergent convergence characteristics as shown in Fig. 2, rendering unified estimation impractical. Consequently, most current transferability estimation methods underperform, resulting in significant drops in performance.

While recent dynamic TE approaches attempt to simulate the evolution of embedding spaces during fine-tuning, they are hindered by efficiency and computational cost constraints. As a result, these methods often depend on manually defined rules to guide feature evolution. However, this trans-

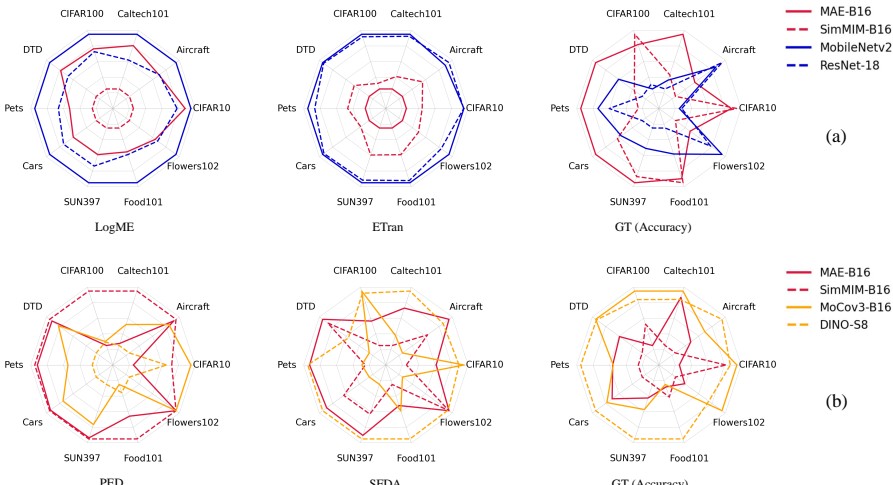

Figure 1: Relative estimated results of recent TE methods across ten benchmarks: (a) Inconsistent estimations between different architectures, such as ViTs and CNNs. (b) Estimation failures across different pre-training strategies, including Instance Discrimination and Masked Image Modeling.

formation process is highly complex and requires sophisticated adaptation to downstream datasets, making accurate modeling challenging under limited computational resources. Thus, these methods frequently fail to predict the transferability of emerging models. Therefore, we propose employing a learnable module to implicitly model the transferability of each unique model by learning the mapping of embedding spaces before and after fine-tuning, thereby enhancing the generalization of transferability estimation (TE).

However, achieving this presents two major challenges. First, the strategy of implicit transferability modeling remains largely unexplored in the TE context, as the evolution process varies across models and downstream tasks, complicating implementation under limited computational resources. Second, the implicit modeling process requires the final embedding states of the model after fine-tuning, which lacks generalizability and is impractical for TE.

To overcome these limitations, we propose an efficient Implicit Transferability Modeling (ITM) paradigm. Instead of modeling the entire evolution of the embedding space, we decouple the intrinsic properties of the models and represent them using an implicit latent representation. We then introduce a Divide-and-Conquer Adaptation (DCA) process to reduce learning complexity and enhance efficiency. To eliminate the need for extensive fine-tuning to obtain the final state, we employ a Pseudo-Clustering-based Optimization (PCO) strategy combined with static and dynamic constraints, allowing for effective estimation without intensive training. By integrating these components, ITM offers more accurate transferability estimation across pre-trained models with diverse architectures and pre-training strategies, significantly outperforming state-of-the-art methods with notable improvements across ten widely used benchmarks, including evaluations of recent pre-trained models.

Our contributions can be summarized as follows:

1. We propose an Implicit Transferability Modeling (ITM) paradigm that incorporates a lightweight learnable mapping module and an efficient evolution process to enhance the precision of embedding space modeling.

2. We introduce a Divide-and-Conquer Adaptation (DCA) strategy, combined with gradient descent acceleration and Pseudo-Clustering Optimization (PCO), to effectively model embedding evolution while minimizing computational costs.

3. We achieve state-of-the-art performance in transferability estimation using ten recent models with diverse architectures from various pre-training methods across ten datasets, achieving an average gain of 16% in rank correlation and significantly outperforming existing methods.

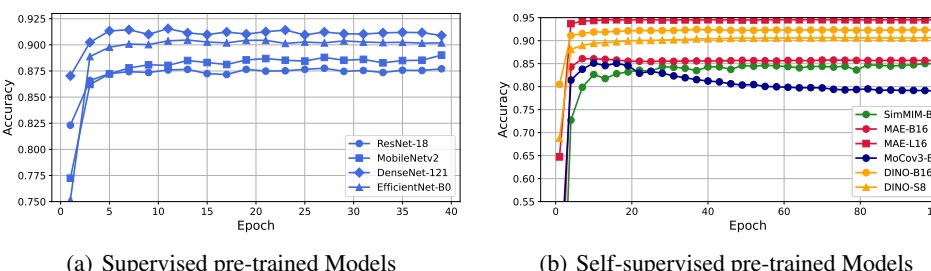

(a) Supervised pre-trained Models        (b) Self-supervised pre-trained Models

Figure 2: The full fine-tuning process of different pre-training methods on the Pets (Parkhi et al., 2012) dataset. The same color represents the same pre-training method, while the same marker denotes models with the same architecture. The training process of different models with supervised pre-training is relatively stable. In contrast, the same ViT-Base model, when pre-trained using different self-supervised methods, exhibits significant variations in the transfer learning process and accuracy on downstream tasks.

## 2 RELATED WORK

### 2.1 TRANSFERABILITY ESTIMATION

Transferability estimation aims to effectively select the most suitable pre-trained model for downstream tasks from a collection of models. Recent approaches can be broadly categorized based on the type of transferability metric used, *i.e.,* static statistics-based and dynamic evolution-based.

**Static statistics-based methods** predict transferability using fixed measurements of pre-trained models. NCE Tran et al. (2019) and LEEP Nguyen et al. (2020) use conditional probability or Bayesian metrics to estimate logit discrepancies between model outputs and downstream task annotations. $\mathcal{N}$LEEP Li et al. (2021) replaces the output layer of LEEP with a Gaussian Mixture Model (GMM) for better efficiency and calibration. LogME You et al. (2021) uses the logarithm of maximum evidence to provide stable predictions with lower computational costs. ETran Gholami et al. (2023) combines various metrics, introducing an energy-based measure to enhance accuracy, while GBC Pándy et al. (2022) employs the Bhattacharyya coefficient to evaluate class separability in the feature space. These methods do not require updates during fine-tuning, resulting in high computational efficiency. However, their inability to model the fine-tuning process limits their prediction accuracy.

**Dynamic evolution-based methods** simulate the model updating process to estimate its state after fine-tuning. SFDA Shao et al. (2022) uses a linear mapping to project initial features into a Fisher space, iteratively enhancing class separability. PED Li et al. (2023) introduces a potential energy-based update model for predicting the evolved state. LEAD Hu et al. (2024) employs ordinary differential equations and downstream objectives to better capture the logits' evolution during adaptation.

Despite their advancements, the recent surge of pre-trained models with varied architectures introduces greater discrepancies in initial states and convergence behaviors (as shown in Fig. 2), challenging these handcrafted prediction-based methods.

### 2.2 PRE-TRAINED MODELS

The Pre-trained and Fine-tuning paradigm in computer vision is widely used to adapt models to specific downstream tasks by leveraging representational knowledge from large-scale pre-training data. Recent research has explored various network architectures and pre-training strategies, leading to significant advancements and a diverse range of model capabilities.

**Architecture evolution.** Initially, pre-training and fine-tuning paradigms showed effectiveness on traditional convolutional neural networks (CNNs) like ResNet He et al. (2016) and DenseNet Huang et al. (2017). Compared to training from scratch, this paradigm helped CNN-based methods achieve superior performance on downstream tasks. More recently, Vision Transformers (ViTs)Dosovitskiy

et al. (2020) and Swin-TransformersLiu et al. (2021) have successfully incorporated self-attention mechanisms (Vaswani et al., 2017) in computer vision. Benefiting from their ability to capture long-range dependencies and global context, ViTs significantly enhance the performance of pre-trained models in tackling complex visual recognition challenges.

**Pre-training strategy.** Early pre-training processes are conducted in a fully supervised manner, with ImageNet Deng et al. (2009) being the most widely used dataset to boost model performance. However, this approach relies heavily on dense human annotations, limiting its broader applicability. With the advent of contrastive learning, strategies like SimCLR (Chen et al., 2020a;b) and MoCo (He et al., 2020; Chen et al., 2020c) show remarkable success in transfer learning. By eliminating the need for ground-truth annotations, these methods significantly increase the amount of available data and outperform traditional supervised pre-training on downstream tasks. Additionally, masked image modeling techniques, such as MAE (He et al., 2022) and SimMIM (Xie et al., 2022), further advance performance, delivering state-of-the-art results across various visual tasks.

However, with diverse pre-training methods and varying model architectures, differences in their optimization processes and feature distributions present greater challenges for existing transferability estimation methods.

## 3 METHODS

### 3.1 PROBLEM FORMULATION

The goal of transferability estimation is to predict the performance ranking of these models after transfer learning on a downstream dataset $\mathcal{D} = \{(x_i, y_i)\}$, which contains $C$ classes. Consider a model zoo $\{\phi_i\}_{i=1}^{M}$, where the models are pre-trained using different architectures and pre-training methods. The true performance $t_i$ of a model $\phi_i$ is obtained by training the model on the training set $\mathcal{D}_T$ and evaluating it on the test set $\mathcal{D}_E$. fine-tuning all candidate models on the dataset requires significant computational resources and time. Therefore, transferability estimation methods aim to output a metric score $s_i$ for each model $\phi_i$ at a low cost and within a short time, with the expectation that this metric will yield a consistent ranking with the true performance ranking after full fine-tuning, thus facilitating effective model selection for downstream applications.

Recent synamic TE methods aim to establish an accurate mapping $\Gamma(\phi, \mathcal{D}_T) : \boldsymbol{E} \to \hat{\boldsymbol{E}}$, where $\boldsymbol{E}$ represents the original embedding space produced by the pre-trained model $\phi$ on the downstream training data $\mathcal{D}_T$, and $\hat{\boldsymbol{E}}$ represents the final state after fine-tuning. However, due to diverse architecture designs and pre-training strategies, models exhibit varying convergence characteristics, making it infeasible to directly model this mapping and generalize across different pre-trained models under affordable computational constraints.

Instead of directly modeling $\Gamma(\cdot, \cdot)$, we propose learning an implicit representation $z$ for the transferability of $\phi$. This approach allows the transfer learning process to be treated as an interaction between the model's transferability and downstream tasks, leading to more effective and adaptable estimation across a broader range of scenarios.

To evaluate the ranking consistency between the predicted metrics $\mathcal{S} = \{s_i\}_{i=1}^{M}$ and the true performance $\mathcal{R} = \{r_i\}_{i=1}^{M}$, a set of metrics such as Spearman's $\rho$(Spearman, 1987), Kendall's $\tau$(Kendall, 1938), and weighted Kendall's $\tau_w$ (Vigna, 2015) can be considered. Following previous work Hu et al. (2024); Li et al. (2023); You et al. (2021), we use weighted Kendall's $\tau_w$ as the primary evaluation metric:

$$\tau_w = \frac{1}{\sum_{i<j} w_{ij}} \cdot \sum_{i<j} w_{ij} \cdot \text{sign}[(G_i - G_j)(P_i - P_j)], \qquad (1)$$

where $G_i \in [1, M]$ and $P_i \in [1, M]$ indicate the ranking of the $i$-th element in $\mathcal{R}$ and $\mathcal{S}$ respectively, and $w_{ij} = \frac{1}{G_i + G_j}$ is the weight accroding to the importance of the model pair $(\phi_i, \phi_j)$.

## 3.2 DIVIDE-AND-CONQUER ADAPTATION

To effectively model the fine-tuning process during transfer learning, we propose a Divide-and-Conquer Adaptation (DCA) strategy that approximates the embedding space evolution using implicit modeling in a computationally efficient manner.

**Embedding space division.** Given the initial embedding space $\boldsymbol{E}$, a module parameterized by $\psi$ aims to approximate the ideal mapping $\Gamma(\cdot, \cdot)$ by maximizing the posterior distribution $q_\psi(\mathrm{z}|\boldsymbol{E})$, where z represents the latent variable capturing intrinsic characteristics of the pre-trained model $\phi$. To achieve an effective approximation of the ideal mapping $\psi(\cdot, \cdot)$, we decompose it as a joint posterior over a set $\mathcal{A}$ containing $K$ independent subspaces sampled from $\boldsymbol{E}$:

$$q_\psi(\mathrm{z}|\boldsymbol{E}) = \sum q_\psi(\mathrm{z}|\boldsymbol{E}_\mathcal{A})p(\mathcal{A}). \tag{2}$$

Based on bayesian decomposition, this posterior can be further refined to factorize across subspaces:

$$q_\psi(\mathrm{z}|\boldsymbol{E}_\mathcal{A}) = \frac{\prod_{j=1}^K p(\boldsymbol{E}_j|\mathrm{z}) \cdot p(\mathrm{z})}{p(\boldsymbol{E}_\mathcal{A})} = \frac{\prod_{j=1}^K \left( \frac{q_\psi(\mathrm{z}|\boldsymbol{E}_j) \cdot p(\boldsymbol{E}_j)}{p(\mathrm{z})} \right) \cdot p(\mathrm{z})}{p(\boldsymbol{E}_\mathcal{A})} = \prod_{j=1}^K q_\psi(\mathrm{z}|\boldsymbol{E}_j). \tag{3}$$

Hence, the posterior $q_\psi(\mathrm{z}|\boldsymbol{E})$ derived from the original embedding space $\boldsymbol{E}$ can be viewed as the product of a series of posteriors $q_\psi(\mathrm{z}|\boldsymbol{E}_j)$ over divisions of the embedding space, where $j \in \{\boldsymbol{E}_j : j \in \mathcal{A}\}$. Thus, modeling the evolution of the entire embedding space $\boldsymbol{E} \to \hat{\boldsymbol{E}}_i$ can be transformed into modeling the evolution of a set of subspaces $\{\boldsymbol{E}_j \to \hat{\boldsymbol{E}}_j\}_{j=1}^K$. In practice, we define the subspace formed by a mini-batch $\mathcal{D}_B$ of data as $\boldsymbol{E}_j$. With larger datasets, the two batches can be considered approximately independent of one another.

This division reduces overall complexity, making it feasible to model the evolution of each subspace independently and ultimately combine them to create a cohesive and efficient approximation of the entire embedding evolution process.

**Dynamic equation-based conquering.** For the evolution modeling of each subspace $\boldsymbol{E}_j$, a naive approach would be to iteratively optimize $\psi(\cdot)$ through its updating process. However, applying this iterative operation to each subspace would lead to an unmanageable computational costs. To address this, we introduce a dynamic equation-based updating mechanism to accelerate convergence for each subspace. To be specific, we model $\psi$ as containing a linear transformation parameterized by weights $\boldsymbol{W}$ for batch-wise mapping. We formulate the objective of the estimation between the predicted embedding $\boldsymbol{E}_j^{(n)}$ at the $n$-th iteration as an L2-norm objective, as follows:

$$\mathcal{L}(\boldsymbol{W}) = \frac{1}{2}||\boldsymbol{E}_j^{(n)}\boldsymbol{W}^{(n)} - \hat{\boldsymbol{E}}_j||_2^2 = \mathrm{Tr}((\boldsymbol{E}_j^{(n)}\boldsymbol{W}^{(n)} - \hat{\boldsymbol{E}}_j)^T(\boldsymbol{E}_j^{(n)}\boldsymbol{W}^{(n)} - \hat{\boldsymbol{E}}_j)). \tag{4}$$

Thus, the updating process based on gradient descent can be formulated as:

$$\begin{cases} \frac{\partial \mathcal{L}}{\partial \boldsymbol{W}^{(n)}} = \boldsymbol{E}_j^{(n)^T}\boldsymbol{E}_j^{(n)}\boldsymbol{W}^{(n)} - \boldsymbol{E}_j^{(n)^T}\hat{\boldsymbol{E}}_j \\ \boldsymbol{W}^{(n+1)} = \boldsymbol{W}^{(n)} - \eta \cdot \frac{\partial \mathcal{L}}{\partial \boldsymbol{W}^{(n)}}, \end{cases} \tag{5}$$

where $\eta$ denotes the learning rate. The evolution of the subspace $\boldsymbol{E}_j$ can be further deduced as:

$$\boldsymbol{E}_j^{(n+1)} = \boldsymbol{E}_j^{(n)} - \eta \cdot \boldsymbol{E}_j^{(n)} \cdot \frac{\partial \mathcal{L}}{\partial \boldsymbol{W}^{(n)}} = (\boldsymbol{I} - \eta\boldsymbol{E}_j^{(n)}\boldsymbol{E}_j^{(n)^T})\boldsymbol{E}_j^{(n)}\boldsymbol{W}^n + \eta\boldsymbol{E}_j^{(n)}\boldsymbol{E}_j^{(n)^T}\hat{\boldsymbol{E}}_j. \tag{6}$$

Denoting $\boldsymbol{C} = \eta\boldsymbol{E}_j^{(n)}\boldsymbol{E}_j^{(n)^T}$, the recursive form of the estimated $\boldsymbol{E}_j^{(n)}$ is finally given by:

$$\boldsymbol{E}_j^{(n+1)} = (\boldsymbol{I} - \boldsymbol{C})\boldsymbol{E}_j^{(n)} + \boldsymbol{C}\hat{\boldsymbol{E}}_j. \tag{7}$$

This recursive formulation efficiently models the evolution of each subspace with reduced computational complexity, ensuring convergence with fewer iterations.

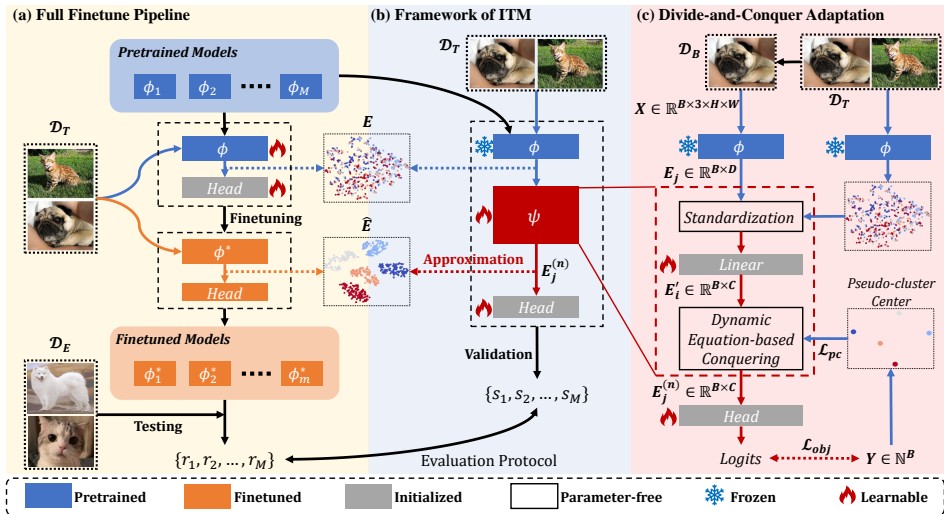

Figure 3: Illustration of full fine-tune and our Improved Transferability Estimation (ITM) approach. (a) Pipeline of full fine-tune. (b) Overview of ITM framework, which is to model the embedding space evolution. (c) Pipeline of DCA on one mini-batch, which is to map the subspace $E_j \to \hat{E}_j$ formed by a batch of data $\mathcal{D}_B$.

### 3.3 PSEUDO-CLUSTERING OPTIMIZATION

During the estimation process, obtaining the final state of the embedding space $\hat{E}_j$ is impractical, as it requires a full fine-tuning process on the downstream task. To eliminate reliance on this final state, we leverage findings indicating that recent pre-trained models possess adequate capability to converge well on the training set (Goodfellow, 2016; Arpit et al., 2017). From this, we draw two inferences: *i.e.,* distribution separability and convergence stability.

First, the static distribution of each class in the embedding space should be well-separated due to optimal convergence on the training set. This implies that the representations of different classes should form distinct clusters. Second, the downstream objectives should converge to a stable point, yielding a minimal loss value. This means the model will effectively align the learned features with the target labels, reducing the error to a minimal level. Based on these two inferences, we utilize pseudo-cluster center constraints and objective-driven updating to approximate the evolution of the embedding space without requiring the actual final state.

**Pseudo-cluster center constraints.** To mimic the static distribution separability of the evolved embedding space, we utilize a pseudo-cluster center generation strategy and introduce a constraint, denoted by MSE loss $\mathcal{L}_{pc}$, that ensures the embeddings of input data are close to their corresponding cluster centers. Specifically, the pseudo-cluster centers can be generated using one-hot or random vectors from high-dimensional embedding spaces or derived from eigenvectors obtained through Principal Component Analysis (PCA) or Laplacian methods for a sparser distribution. In the experimental section, we compare the impact of different generation methods and determine the optimal approach based on empirical results.

**Objective-driven updating.** To achieve convergence stability, we incorporate downstream task objectives $\mathcal{L}_{obj}$ into the estimation of $\psi$, ensuring that the iterative adaptation aligns closely with downstream requirements. During this process, the downstream objective is updated iteratively to achieve stable convergence in the logit space, ultimately guiding the embedding evolution toward effective feature alignment.

### 3.4 FRAMEWORK

Combining the proposed Divide-and-Conquer Adaptation (DCA) and Pseudo-Cluster Optimization (PCO) modules, we form the ITM framework. The overall framework is illustrated in Fig. 3. During

the estimation process, a lightweight module $\psi$ is introduced to learn the implicit transferability of the given model. By interacting with downstream data through DCA and optimizing using PCO, this module effectively captures the transferability potential and adapts efficiently, facilitating precise transferability estimation while maintaining computational efficiency. The final objective can be formulated as follows:

$$\mathcal{L} = \mathcal{L}_{obj} + \lambda \mathcal{L}_{pc}, \tag{8}$$

where $\lambda$ is a hyperparameter that balances these two loss components and is controlled by $C$ with $\eta$ in equation 7. To further enhance the proposed framework, we adopt embedding pre-Standardization and a dynamic learning Rate adjustment strategy and to facilitate efficient convergence during optimization.

**Embedding pre-standardization** Embedding pre-standardization is also applied to ensure that the initial embedding space $E$ has a consistent scale and distribution, which prevents gradients from exploding or vanishing and promotes more stable updates during the iterative adaptation process.

**Dynamic learning rate adjustment.** Since $n, \eta$ are critical hyperparameters in DCA that determine the accelerated process, we aim to reduce the dependency on fixed values and avoid manually tuning $n$ and $\eta$ for each different pre-trained model and downstream task. Therefore, we introduce a dynamic adjustment mechanism where the learning rate $\eta$ is adaptively scaled according to the standardized initial state $E$ and set $n$ constant to 1. This allows the framework to automatically adjust $\eta$ for optimal convergence speed and stability, providing more flexibility across diverse scenarios.

By combining these strategies, the framework achieves more consistent and efficient convergence across various pre-trained models and downstream tasks.

## 4 EXPERIMENTS

We constructed a benchmark using 10 classic single-label image classification datasets and 10 different pre-trained models. We then reproduced several of the best existing methods for transferability estimation, highlighting the superior performance of our proposed approach. Subsequently, we conducted ablation studies to analyze the contributions of each component in our method.

### 4.1 BENCHMARK

**Downstream datasets.** We utilized 10 commonly used single-label image classification datasets in transfer learning, including CIFAR-10, CIFAR-100 (Krizhevsky et al., 2009), FGVC Aircraft (Maji et al., 2013), Caltech-101 (Fei-Fei et al., 2004), DTD (Cimpoi et al., 2014), Oxford-IIIT Pets (Parkhi et al., 2012), Stanford Cars (Krause et al., 2013), SUN-397 (Xiao et al., 2010), Food-101 (Bossard et al., 2014), and Oxford102 Flowers (Nilsback & Zisserman, 2008). These datasets encompass image classification tasks across different scenarios, featuring varying numbers of categories and dataset sizes.

**Pre-trained model zoo.** To ensure applicability in real-world scenarios, we selected 10 models that vary in pre-training methods and architectural designs. Specifically, for models pre-trained using supervised learning, we employed ResNet-18 (He et al., 2016), MobileNetv2 (Sandler et al., 2018), EfficientNet-B0 (Tan, 2019), and Densenet-121 (Huang et al., 2017). For contrastive learning-based models, we included DINO-S8 (Caron et al., 2021), DINO-B16 (Caron et al., 2021), and MoCov3-B16 (Chen et al., 2021). Lastly, for models pre-trained with Masked Image Modeling (MIM), we utilized MAE-B16 (He et al., 2022), MAE-L16 (He et al., 2022), and SimMIM-B16 (Xie et al., 2022). In this context, S/B/L represent the small, base, and large versions of the ViT (Dosovitskiy et al., 2020) model, while 8/16 indicate the patch sizes used in the ViT architecture.

**Ground truth model rank.** For all models, we added a classification head to facilitate transfer learning on downstream datasets. We used the AdamW (Loshchilov, 2017) optimizer to train both the pre-trained backbone and the randomly initialized classification head. The grid search strategy specifically involves exploring a range of learning rates from the set $\{10^{-5}, 2 \times 10^{-5}, 5 \times 10^{-5}\}$ and weight decay values from the set $\{10^{-2}, 10^{-4}\}$. At the end of each epoch, we evaluated the model on the test set. Ultimately, we recorded the highest test accuracy achieved by each model across multiple experiments for each dataset as the model's accuracy on that dataset.

Table 1: Comparison of weighted Kendall's $\tau_w$ and wall-Clock Time (s) for different methods on various datasets. We reproduced methods for comparison including PED (Li et al., 2023), LogME (You et al., 2021), $\mathcal{N}$Leep (Li et al., 2021), PARC (Bolya et al., 2021), SFDA (Shao et al., 2022) and Etran (Gholami et al., 2023). All methods are run and timed on the same CPU environment.

| Methods | Cal101 | Cars | CIFAR100 | CIFAR10 | DTD | Aircraft | Flowers | Food | Pets | SUN | Avg. |
|---|---|---|---|---|---|---|---|---|---|---|---|
| weighted Kendall's $\tau_w$ ↑ | | | | | | | | | | | |
| PED | 0.32 | -0.01 | 0.51 | 0.77 | 0.06 | -0.20 | 0.16 | **0.60** | -0.20 | 0.07 | 0.21 |
| LogME | **0.71** | 0.36 | 0.56 | 0.61 | 0.61 | 0.22 | **0.77** | 0.15 | 0.14 | 0.38 | 0.45 |
| $\mathcal{N}$Leep | 0.47 | 0.04 | 0.32 | 0.48 | 0.57 | 0.13 | 0.62 | 0.24 | 0.30 | 0.01 | 0.32 |
| PARC | 0.08 | 0.00 | -0.07 | 0.25 | 0.42 | 0.12 | 0.62 | 0.19 | 0.10 | 0.01 | 0.17 |
| SFDA | 0.59 | 0.07 | 0.48 | **0.79** | 0.13 | 0.18 | -0.39 | 0.33 | 0.28 | 0.09 | 0.25 |
| ETran | 0.13 | -0.06 | -0.14 | 0.21 | 0.36 | 0.27 | 0.08 | 0.23 | 0.38 | -0.06 | 0.14 |
| ITM (Ours) | 0.56 | **0.61** | **0.59** | 0.69 | **0.77** | **0.43** | 0.65 | 0.44 | **0.73** | **0.62** | **0.61** |
| Wall-Clock Time (s)) ↓ | | | | | | | | | | | |
| PED | 6.99 | 8.31 | 34.32 | 46.80 | 2.33 | 5.95 | 2.80 | 41.72 | 3.94 | 8.29 | 16.14 |
| LogME | 0.85 | 1.32 | 4.50 | 2.62 | 0.50 | 0.86 | 0.54 | 6.29 | 0.53 | 1.31 | 1.93 |
| $\mathcal{N}$Leep | 25.52 | 44.91 | 862.49 | 268.25 | 4.60 | 17.65 | 3.42 | 1387.65 | 4.31 | 47.33 | 266.61 |
| PARC | 14.42 | 19.82 | 116.65 | 118.19 | 0.74 | 13.14 | 0.25 | 106.19 | 3.53 | 19.80 | 41.27 |
| SFDA | 4.02 | 5.43 | 20.11 | 18.56 | 1.70 | 3.92 | 1.91 | 28.86 | 2.20 | 5.43 | 9.21 |
| ETran | 1.71 | 2.30 | 7.58 | 6.88 | 0.77 | 1.64 | 0.98 | 10.63 | 0.96 | 2.31 | 3.58 |
| ITM (Ours) | 7.50 | 8.50 | 9.40 | 7.90 | 7.00 | 7.50 | 7.20 | 10.50 | 7.20 | 11.50 | 8.42 |

## 4.2 IMPLEMENTATION DETAILS

In the default experimental setup for ITM, we set the number of training iterations to only 500, performing a test on the validation set every 100 iterations. In our experiments, ITM, similar to other transferability estimation methods, utilizes only the data from the original training set $\mathcal{D}_T$. And $\mathcal{D}_T$ is randomly divided in a 4:1 ratio for training and validation purposes in ITM. As described in Sec. 3.4, the default configuration for DCA includes $n = 1$ and an adaptive $\eta$. For all experiments, we use a learning rate of $5 \times 10^{-3}$ along with the AdamW optimizer. Additional experimental settings can be found in the appendix.

## 4.3 COMPARISON WITH PREVIOUS APPROACHES

We conducted a comprehensive comparison of our method against previous transferability estimation methods on the benchmark described in Sec. 4.1. This comparison includes the primary weighted Kendall's $\tau_w$ as well as the wall-clock time.

As shown in Table 1, when incorporating the diversity of pre-trained models, our benchmark presents challenges to the generalization of TE methods. The average performance of most TE methods has deteriorated. Our method ITM achieved an average performance of 0.61 in terms of the weighted Kendall's $\tau_w$ across the 10 datasets, outperforming the next-best method, LogME, by 16%. Specifically, ITM achieved the highest weighted Kendall's $\tau_w$ on six datasets and the second-highest on two classification datasets. Although other methods perform well on certain specific datasets, it is evident that they nearly "fail" on some datasets. The lowest weighted Kendall's $\tau_w$ correlation coefficients for these methods are -0.2, 0.14, 0.01, -0.07, -0.39, and -0.14, respectively, while ITM achieves a minimum of 0.43 even on its worst-performing dataset. This highlights that ITM offers significantly better stability compared to previous methods, which is particularly important for practical applications.

In terms of runtime, many effective transferability estimation methods are significantly faster than pre-trained models, which require 738 seconds for feature extraction. As shown in the table, although ITM uses backpropagation for parameter updates like common deep learning models, it contains only one linear layer in the DCA. As a result, ITM achieves a runtime of just 8.42 seconds in a CPU environment, demonstrating an excellent trade-off between speed and accuracy.

## 4.4 ABLATION STUDY

### 4.4.1 BATCH SIZE

As described in Sec. 3.2, the DCA module operates on a subspace formed by a single batch. Therefore, we adjusted the batch size $B$ and illustrated its impact on accuracy and runtime in Fig. 4. As

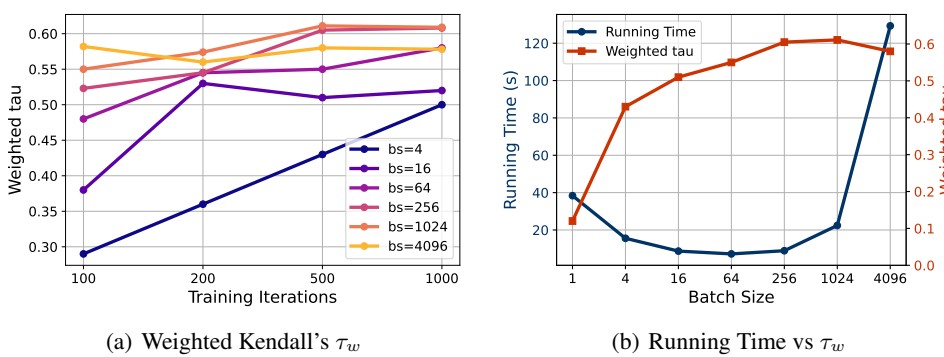

(a) Weighted Kendall's $\tau_w$        (b) Running Time vs $\tau_w$

Figure 4: Quantitative Ablation Experiment on Batch Size in DCA. (a) Weighted Kendall's $\tau_w$ as a function of batch size and the number of training iterations. (b) Running time and $\tau_w$ of DCA for 500 iterations as a function of batch size. All experiments are conducted in a CPU environment.

Table 2: Quantitative ablation study on the feature standardization and adaptive learning rate. We calculated the average weighted Kendall's $\tau_w$ coefficient for each experiment after training for 100, 200, 500, and 1000 iterations.

| Standardization | Adaptive $\eta'$ | Average weighted Kendall's $\tau_w$ | | | |
|:---:|:---:|:---:|:---:|:---:|:---:|
| | | 100 | 200 | 500 | 1000 |
| × | × | 0.443 | 0.482 | 0.519 | 0.541 |
| × | ✓ | 0.471 | 0.496 | 0.508 | 0.532 |
| ✓ | × | 0.557 | 0.534 | 0.576 | 0.569 |
| ✓ | ✓ | 0.523 | 0.545 | 0.605 | 0.608 |

shown in the figure, when the batch size is less than or equal to 1024, the accuracy and convergence speed of ITM improve with increasing batch size. However, when the batch size is further increased to 4096, ITM quickly overfits, and its weighted Kendall's $\tau_w$ fluctuates across training iterations. In Fig. 4(b), we compare the running time of ITM with its weighted Kendall's $\tau_w$ after 500 training iterations. Since the training and validation batch sizes must remain consistent, ITM runs slightly slower when the batch size is small due to the smaller testing batch size. However, when the batch size is greater than or equal to 1024, the speed of ITM significantly decreases because the time complexity of Equation 7 is $O(B^3)$. Considering both speed and accuracy, we set the default batch size for ITM to 256 and the number of training iterations to 500.

### 4.4.2 STANDARDIZATION AND ADAPTIVE LEARING RATE

We conducted ablation experiments on the standardization and the adaptive learning rate mentioned in Sec. 3.4. As shown in Tab. 2, the adaptive learning rate does not significantly improve ITM when standardization is not applied, as the adaptive learning rate is derived based on the standardized embedding space. However, as indicated in the third row of the table, standardizing the features greatly enhances the convergence speed and accuracy of ITM. This improvement occurs because the distribution of features from different models varies significantly, and standardization helps the model converge more quickly. After incorporating both standardization and the adaptive learning rate, ITM achieved the highest performance. This indicates that setting different learning rates for various models based on the initial intra-class dispersion of the standardized model features significantly aids ITM in mapping the embedded spaces before and after fine-tuning.

### 4.4.3 PSEUDO-CLUSTER CENTERS

As shown in Fig. 5, we attempted to generate orthogonal pseudo-cluster centers on the unit sphere using different methods as the target for feature alignment. Here, "PCA" and "Laplacian" refer to the orthogonalization methods applied after random generation using PCA and the Laplacian

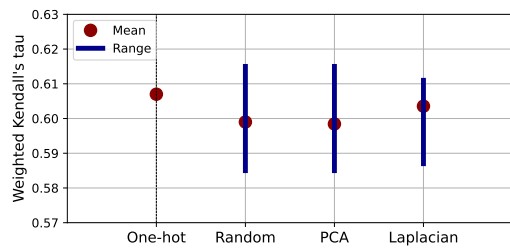

Figure 5: The performance of different methods for generating orthogonal pseudo-cluster centers under 10 repeated experiments.

matrix, respectively. Through repeated experiments, we found that the performance of the various methods showed little difference. This is because the classification head $h$ performs an additional linear transformation on the embedded features, making the choice of pseudo-cluster centers robust. To eliminate the random instability in practical usage, we adopted one-hot encoding of the classes as the centers.

## 5 CONCLUSIONS

In this work, we propose ITM, an Improved Transferability Modeling paradigm designed to decouple the intrinsic properties of pre-trained models by representing them in an implicit latent space, providing an efficient solution for assessing model transferability. We introduce a Divide-and-Conquer Adaptation process to reduce learning complexity and computational costs, and employ a Pseudo-Clustering-based Optimization strategy that eliminates the need for extensive fine-tuning, enabling effective estimation without intensive training. We conduct experiments on recent models and a diverse set of downstream tasks, showing that the proposed ITM significantly outperforms all counterparts with superior computational efficiency, demonstrating its effectiveness.

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

# A APPENDIX

## A.1 BENCHMARK

As shown in Tab. 3, we present the performance of ten different pre-trained models on ten downstream single-label classification datasets.

Table 3: The highest accuracy of different models across various datasets.

| Models | Cal101 | Cars | CIFAR100 | CIFAR10 | DTD | Aircraft | Flowers | Food | Pets | SUN |
|---|---|---|---|---|---|---|---|---|---|---|
| DenseNet-121 | 97.24 | 88.38 | 85.67 | 97.38 | 69.47 | 83.86 | 90.40 | 84.99 | 91.77 | 72.49 |
| EfficientNet-B0 | 97.87 | 87.25 | 87.01 | 97.88 | 68.88 | 81.61 | 89.71 | 85.82 | 90.68 | 73.87 |
| MobileNetv2 | 96.03 | 85.65 | 82.15 | 96.48 | 67.34 | 76.36 | 90.03 | 83.70 | 89.02 | 71.16 |
| ResNet-18 | 95.74 | 83.58 | 82.69 | 96.57 | 65.59 | 77.56 | 88.52 | 79.92 | 88.50 | 68.71 |
| DINO-B16 | 98.16 | 88.42 | 90.53 | 98.70 | 74.31 | 78.40 | 93.19 | 87.88 | 92.97 | 77.02 |
| DINO-S8 | 97.47 | 89.33 | 90.24 | 98.65 | 71.97 | 80.02 | 90.84 | 90.84 | 93.10 | 77.53 |
| MoCov3-B16 | 97.70 | 88.53 | 90.74 | 98.68 | 72.07 | 75.70 | 93.61 | 87.15 | 89.75 | 75.92 |
| MAE-B16 | 97.52 | 88.16 | 87.55 | 98.42 | 69.04 | 72.16 | 85.80 | 87.30 | 89.81 | 75.29 |
| MAE-L16 | 97.98 | 91.21 | 91.28 | 98.55 | 74.26 | 85.30 | 90.73 | 90.82 | 94.69 | 79.18 |
| SimMIM-B16 | 96.20 | 86.02 | 88.80 | 98.63 | 66.17 | 68.32 | 83.87 | 87.88 | 87.16 | 74.55 |

## A.2 OTHER EVALUATION PROTOCOLS

We evaluated the transferability estimation methods based on other rank correlation metrics, including Spearman's $\rho$ (Spearman, 1987) and Kendall's $\tau$ (Kendall, 1938). As shown in Tab. 4, ITM also achieved the best performance on these two metrics.

Table 4: Comparison of Spearman's $\rho$ (Spearman, 1987) and Kendall's $\tau_w$ (Kendall, 1938) for different methods on various datasets.

| Methods | Cal101 | Cars | CIFAR100 | CIFAR10 | DTD | Aircraft | Flowers | Food | Pets | SUN | Avg. |
|---|---|---|---|---|---|---|---|---|---|---|---|
| | | | | | Spearman's $\rho$ | | | | | | |
| PED | 0.61 | 0.05 | 0.81 | 0.92 | 0.19 | -0.37 | -0.07 | 0.84 | -0.18 | 0.25 | 0.31 |
| LogME | 0.81 | 0.61 | 0.62 | 0.61 | 0.77 | 0.45 | 0.90 | 0.20 | 0.37 | 0.52 | 0.59 |
| $\mathcal{N}$Leep | 0.37 | 0.24 | 0.25 | 0.42 | 0.53 | 0.44 | 0.79 | 0.09 | 0.41 | 0.01 | 0.35 |
| PARC | -0.01 | 0.10 | -0.19 | 0.05 | 0.49 | 0.43 | 0.79 | 0.03 | 0.26 | -0.05 | 0.19 |
| SFDA | 0.71 | 0.16 | 0.68 | 0.70 | 0.09 | 0.38 | -0.58 | 0.43 | 0.52 | 0.13 | 0.32 |
| ETran | -0.01 | -0.08 | -0.28 | 0.01 | 0.09 | 0.54 | 0.13 | 0.05 | 0.39 | -0.13 | 0.07 |
| ITM (Ours) | 0.75 | 0.84 | 0.71 | 0.70 | 0.87 | 0.54 | 0.78 | 0.45 | 0.84 | 0.75 | 0.72 |
| | | | | | Kendall's $\tau$ | | | | | | |
| PED | 0.556 | 0.067 | 0.644 | 0.778 | 0.111 | -0.333 | -0.061 | 0.689 | -0.156 | 0.111 | 0.241 |
| LogME | 0.600 | 0.422 | 0.422 | 0.467 | 0.600 | 0.333 | 0.778 | 0.111 | 0.244 | 0.378 | 0.436 |
| $\mathcal{N}$Leep | 0.289 | 0.156 | 0.244 | 0.289 | 0.422 | 0.289 | 0.600 | 0.156 | 0.333 | 0.067 | 0.284 |
| PARC | 0.022 | 0.111 | -0.156 | 0.022 | 0.378 | 0.289 | 0.600 | 0.067 | 0.200 | 0.067 | 0.160 |
| SFDA | 0.511 | 0.156 | 0.467 | 0.600 | 0.067 | 0.244 | -0.479 | 0.244 | 0.422 | 0.111 | 0.234 |
| ETran | -0.022 | -0.022 | -0.156 | -0.022 | 0.111 | 0.422 | 0.111 | 0.156 | 0.378 | -0.067 | 0.089 |
| ITM (Ours) | 0.629 | 0.644 | 0.556 | 0.600 | 0.719 | 0.422 | 0.600 | 0.378 | 0.719 | 0.600 | 0.587 |

## A.3 ADAPTIVE LEARNING RATE

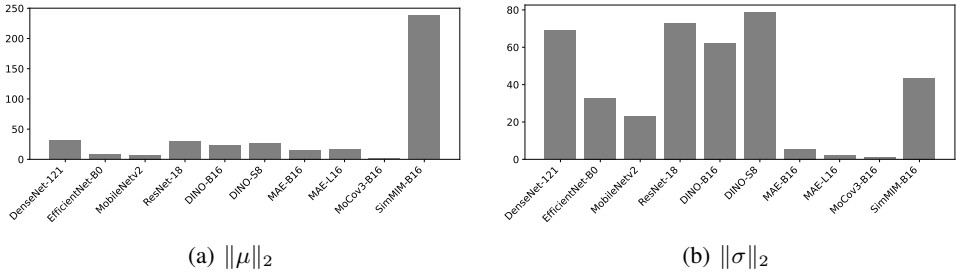

(a) $\|\mu\|_2$        (b) $\|\sigma\|_2$

Figure 6: The mean and variance of features from different pre-trained models on Cal-101.

The goal of DCA is to mapping the embedding spaces of the pre-trained backbone before and after full fine-tuning. However, as shown in Fig. 6, different pre-trained models have initial embedding spaces with different means and variances, which means that using the same $\eta$ in equation 7 may be unfair to models with initially poor feature distributions. Since equation 7 is similar to a linear recurrence, we computed the average Euclidean distance of all initial features in initial embedding space $\boldsymbol{E} = \{f_i\}_i^n$ of model $\phi$, after standardization, to the cluster centers of their respective classes: $\text{dis}_\phi = \frac{1}{n}\sum_{j=1}^n \|f_i' - \bar{f}'_{y_i}\|_2$ where $f_i' = \frac{f_i - \bar{f}}{\sigma(f)}$ represents the standardized features, and $\bar{f}'_c$ denotes the mean of the features for class $y_i = c$.

Since the linear recurrence coefficient $\boldsymbol{C} = \eta \boldsymbol{E}_j \boldsymbol{E}_j^T$ is complex, we consider the recurrence relation when batch size $B = 1$, $n = 1$, and $\boldsymbol{C} = \eta$:

$$\boldsymbol{E}_i^{(n)} = (1-\eta)\boldsymbol{E}_i + \eta_i \hat{\boldsymbol{E}}_i, \tag{9}$$

$$\boldsymbol{E}_j^{(n)} = (1-\eta)\boldsymbol{E}_j + \eta_j \hat{\boldsymbol{E}}_j. \tag{10}$$

This represents two initial spaces $\boldsymbol{E}_i$ and $\boldsymbol{E}_j$ being updated towards the target $\hat{\boldsymbol{E}}_i$ and $\hat{\boldsymbol{E}}_j$ using different linear recurrence coefficients $\eta_i$ and $\eta_j$. By setting $\|\boldsymbol{E}_i^{(n)} - \hat{\boldsymbol{E}}_i\|_2 = \|\boldsymbol{E}_j^{(n)} - \hat{\boldsymbol{E}}_j\|_2$, we obtain the equation:

$$\|(1-\eta_i)\boldsymbol{E}_i + \eta_i\hat{\boldsymbol{E}}_i - \hat{\boldsymbol{E}}_i\|_2 = \|(1-\eta_j)\boldsymbol{E}_j + \eta_j\hat{\boldsymbol{E}}_j - \hat{\boldsymbol{E}}_j\|_2$$
$$\implies (1-\eta_i)\|\boldsymbol{E}_i - \hat{\boldsymbol{E}}_i\|_2 = (1-\eta_j)\|\boldsymbol{E}_j - \hat{\boldsymbol{E}}_j\|_2. \tag{11}$$

This implies that $(1 - \eta)$ is inversely proportional to the Euclidean distance between the features and the target point. Therefore, we compute an adaptive learning rate $\eta$ based on the model's $\text{dis}_\phi$:

$$\eta_i = f(\text{dis}_{\phi_i}) = 1 - \frac{(1-\eta_b)\text{dis}_b}{\text{dis}_{\phi_i}}. \tag{12}$$

In equation 12 we set $\eta = \eta_b$ as an anchor for the model $\phi$ when $\text{dis}_\phi = \text{dis}_b$.

### A.4 Embedding Space Mapping

As shown in the figure, the embedding distributions of pre-trained models $\boldsymbol{E}$ from different pre-training methods are quite disparate, but after fine-tuning, their embedding space $\hat{\boldsymbol{E}}$ converge to a much better distribution. Previous methods either solely rely on pre-trained model features or use crude methods to simulate the fine-tuning process, which explains why these methods perform poorly on such complex benchmarks.

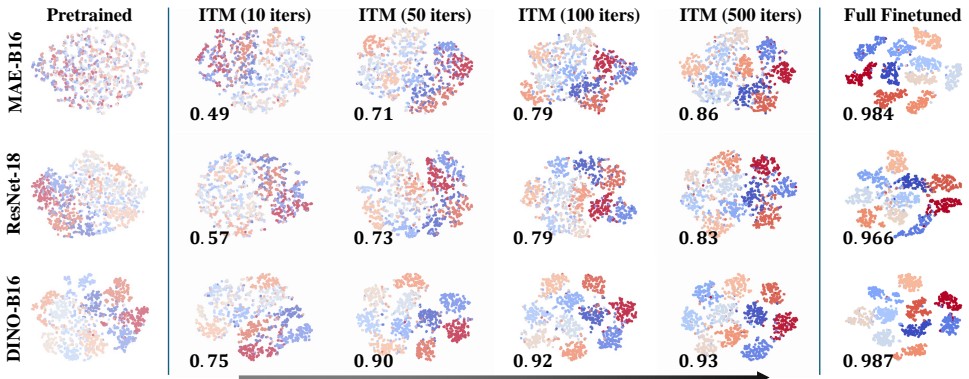

Figure 7: Illustration of the embedding space mapping. The figure includes the embedding distributions of ResNet-18, DINO-B16, and MAE-B16 models on the CIFAR10 test set for both $\boldsymbol{E}$ and $\hat{\boldsymbol{E}}$, as well as the changes in $\boldsymbol{E}_j^{(n)}$ during the ITM process. The visualizations are generated using T-SNE for dimensionality reduction. The numbers in the lower left corner of each image represent the classification accuracy of the model on the test set at that moment.

