# OpenReview forum: "Generalizable Transferability Estimation of Foundation Vision Models via Implicit Learning"
_ICLR.cc/2025/Conference — ICLR 2025 Conference Withdrawn Submission_

### Official Review · Reviewer_kJcc · 2024-10-26

**Soundness:** 3
**Presentation:** 3
**Contribution:** 3
**Rating:** 5
**Confidence:** 4

**Summary:**

This paper tackles the problem of transferability estimation, which aims at selecting the best pretrained model for specific downstream tasks. The authors propose the implicit transferability method to efficiently model the transfer process, reducing both learning complexity and computational costs. Specifically, they propose a divide and conquer adaptation process to model the transfer process. They also introduce a Pseudo-Clustering-based Optimization (PCO) strategy with static and dynamic constraints to estimate transferability without intensive retraining. Their approach outperforms current methods, showing significant improvements across ten benchmarks, making it highly effective and generalizable for model selection.

**Strengths:**

[1] The method tackles the problem of transferability estimation. The author conducts exhaustive methods to validate the effectiveness of the proposed methods.

[2] The method seems to be reasonable.

**Weaknesses:**

[1] The motivation is unclear. In the introduction (L081-085), the author states two challenges, i.e., the strategy of implicit transferability modeling remains largely unexplored and the implicit modeling process requires the final embedding states of the model after fine-tuning. However, it can not fully understand the where the challenge is. The author more specifically explain the challenges.

[2] The description of the method is not clear. For example. In L221-L222, the embedding space division: The author is suggested to briefly introduce the motivation of division, what to divide and how to divide. Currently, I can not fully understand the motivation and implementations.

[3] The pseudo-clustering accuracy is essential to transferability estimation. The authors are suggested to evaluate the sensitivity of the clustering performance.

**Questions:**

See the weakness.

---

### Official Review · Reviewer_dEyf · 2024-10-28

**Soundness:** 2
**Presentation:** 2
**Contribution:** 1
**Rating:** 1
**Confidence:** 4

**Summary:**

In this paper transferability estimation is studied, that is, rank a set of pre-trained networks for a specific target task, using a computational efficient method (the underpinning assumption is that fine tuning all pre-trained networks is too time consuming). In this work, the mapping from the original embedding space (E) of a model from the zoo, to the fine-tuned embedding space (E’) on the target data is estimated via a combination of three loss functions, and this is used to estimate the transferability estimation score. The method is validated on a set of 10 classification tasks datasets, with a model-zoo of 10 models.

**Strengths:**

The paper starts with a reasonably strong analysis of transferability estimation, that these yield inconsistent estimates between different architectures or pre-training methods. This is interesting to see, and yields already some questions. For example, from Fig 1: I mainly observe that LogMe and ETran have a preference of a single network architecture for any target dataset, so their transferability scores seems to be a function of the pre-trained network, more than for the given target dataset. Does that hold for other TE methods as well? Do the target tasks prefer different models, or is there just a single best model for all target tasks? And from Fig 2: this plot makes me wonder what the performance of a supervised ViT model would be, it seems that it is not in the mix of pre-trained models?

The individual elements of the proposed method (estimating target embedding, using some pseudo clustering) seems to make sense.

**Weaknesses:**

# Major weaknesses

1. The biggest weakness is that the method is not clearly presented, while conceptually the elements make sense, the interconnection and the final TE method (or scoring) is not directly clear from the manuscript. To make this weakness more concrete:
    - (major) The final transferability estimation score (s_i) for model i (for a particular target task) is not explicitly defined. It is related to the ideal mapping gamma, but not provided.
    - (major) The loss (eq 7) uses Ê (E-hat), which is defined above Eq 7 as the final embedding state after fine-tuning. Defining a transferability metric which works after fine-tuning is probably not the goal, so please clarify.
    - (major) How is the lambda parameter (Eq 8), controlled by the learning rate (eta) in Eq 7 as stated in L331. How is Eq 8 in general related to Eq 7? Eq 8 defines a joint loss, Eq 7 only an update of the parameters.
    - (major) What is the objective driven loss? How is it defined?
    - (major) In the implementation details it is stated that some of the target dataset is used for validation purposes in ITM. What parts are validated, and when in the pipeline, and how is that used?
    - The difference between mapping Gamma (L222) and mapping Phi (L224) is not clear to me. And what is the difference between psi(.,.)(L224)  and psi(.) (L245)?
    - What is z, it seems only implicitly used in the (approximation) of ^Z (Z-hat)? Moreover, the decomposition of q_phi(z|E), is purely based on the independence assumption of E, I don’t think a double bayes rule is necessary to write that down. Finally, how is q used? Is the loss in Eq 4, a sum over all j in K (of E 3)?
    - What is the difference between the subsets A (indexed by j) and the subset of K (also indexed by j)?
    - What does it mean that the process is “treated as an interaction between the model’s transferability and downstream tasks, leading to more effective and adaptable estimation across a broader range of scenarios.” What interaction is used, is the model learned once for multiple downstream tasks? Does it then generalize to others?

2. The second biggest weakness of the current study is the experimental evaluation, although this is a more general problem within transferability estimation literature, but the current experimental evaluation is too weak to draw any conclusions. The main problem is that a single ranking experiment is evaluated, while the outcome of this ranking really does depend on the selected individual models in the model-zoo. This means that adding or removing a single model to/from the zoo all results are likely to be significantly different. This has been studied in [AgostinelliECCV22]. Therefore, the method should be evaluated on a larger set of ranking experiments. This is not too complicated nor too computational expensive: one could draw samples from a larger model-zoo. For example one could draw all sets of 10 models out of a model-zoo of 14 models, this provides (approx) 1000 ranking experiments (14 choose 10). This only requires computing once the scores (s) and ground truths (r) for 4 additional models, while providing 999 more ranking tests. Without such an experiment, I think it is impossible to draw any conclusion on the success of any method.

# Minor weaknesses
- Figure 1: I’m unsure what I see (and should see) in this plot. It is remarkable that the MAE-B16/SimMIM-B16 models have different GT radar plots between (a) and (b).
- Figure 2: The y-scales are different from plot (a) and plot (b).
- (nitpick) In the related work the `cite` command is used, where it should be the `citep`, for example: NCE Tran et al. (2019) → NCE (Tran et al., 2019).
- (nitpick) The sub-index i is used for the number of elements in the dataset (L186) and for the number of models in the zoo (L187)
- It is unclear how many epochs over the target task training data are performed by each method. Most transferability estimation methods assume (eg) 1 epoch (to get feature embeddings and target labels).
- In the abstract and introduction ‘generalizability’ of a transfer estimation method is mentioned, what do you mean with that?

# References
- [**AgostinelliECCV22**] Agostinelli et al, How stable are Transferability Metrics evaluations?, ECCV 2022.

**Questions:**

The questions are mostly related to the major weakness mentioned:

1. Please run a larger set of ranking experiments, using subsampling from a (larger) model-zoo and share these results, preferably in some scatter plot (eg LogMe vs ITM) on all 10 target datasets.
2. Please specify the final transferability scores s of the ITM method for model i and a target task.
3. Please explain what Ê (E-hat) is in Eq 7 and how it is obtained (without full fine-tuning).
4. Please provide the loss functions used for Lpc and Lobj and how they are used to compute s.
5. Please explain how the validation data is used in the ITM method.
6. Please explain for how many epochs ITM is run over the target data (and how that compares to related work).

---

### Official Review · Reviewer_jmVU · 2024-10-28

**Soundness:** 3
**Presentation:** 2
**Contribution:** 3
**Rating:** 5
**Confidence:** 5

**Summary:**

This paper presents an Implicit Transferability Modeling (ITM) paradigm that employs an implicit modeling strategy to capture the intrinsic properties of pre-trained models. The ITM framework integrates a Divide-and-Conquer Adaptation (DCA) process for efficient modeling of independent subspaces, along with a Pseudo-Clustering-based Optimization (PCO) strategy to enable effective estimation without extensive fine-tuning.

**Strengths:**

1. The study addresses a significant research area: model selection. The ability to choose an appropriate pre-trained model for specific downstream tasks is crucial and has the potential to enhance performance outcomes.

2. Great visualization of the proposed method, providing clear insights into the framework's structure and functionality.

**Weaknesses:**

1. In line 225, the methodology mentions "K independent subspaces sampled from E"; however, the process for obtaining these divisions is not adequately described. The criteria for determining K and the methodology used to derive these independent subspaces require further clarification. It would be helpful if the authors could provide specific examples of how K is determined and the practical sampling process for the subspaces.

2. In line 307, the generation of pseudo-cluster centers is noted to involve various methods, including the use of random vectors from high-dimensional space. This approach appears to conflict with conventional methods (centers are determined by some clustering methods) but lacks sufficient explanation within the text.

3. The evaluation of the proposed method is conducted solely on a single benchmark, omitting more widely recognized benchmarks (used in SFDA[1], Logme[2]), which limits the robustness of the findings.

[1] Not All Models Are Equal: Predicting Model Transferability in a Self-challenging Fisher Space.

[2] LogME: Practical Assessment of Pre-trained Models for Transfer Learning

**Questions:**

1. How is the independence of the K subspaces ensured? If the method involves dimensional grouping, does this assumption of independence hold true in practical scenarios?

2. The random generation of pseudo-cluster centers, as mentioned in Weakness 2, seems counterintuitive. A more thorough explanation would enhance understanding

3. I find it noteworthy that the predicted state in Equation (7) exhibits a similar format to the predictions made in LEAD[3]. Both methodologies utilize dynamic equations to model the evolution process, yielding initial and final state interpolations. It would be beneficial to provide a more detailed comparison between their method and LEAD, highlighting the key differences and potential advantages of their approach. This would help clarify the novelty and contribution of this work.

4. The experimental results indicate that previous dynamic evolution-based methods perform poorly on the proposed benchmark. A detailed explanation, accompanied by visualizations (e.g., Figure 7), illustrating the advantages of the proposed method in modeling the evolution process compared to earlier approaches would strengthen the discussion.

5. The absence of evaluation on common benchmarks (e.g., SFDA, Logme) raises concerns regarding the generalizability of the method. I recommend supplementing the results with additional evaluations on these established benchmarks to demonstrate the method's applicability.

[3] LEAD: Exploring Logit Space Evolution for Model Selection.

---

### Official Review · Reviewer_YmQa · 2024-11-02

**Soundness:** 2
**Presentation:** 3
**Contribution:** 2
**Rating:** 5
**Confidence:** 4

**Summary:**

This paper introduces an Implicit Transferability Modeling (ITM) paradigm to improve the accuracy of embedding space modeling, with experimental validation conducted across multiple datasets. Overall, the paper looks interesting.

**Strengths:**

- Easy to follow
- Tested on multiple datasets

**Weaknesses:**

1. In Lines 240-243 and 268-269, the authors state that 'the division and the recursive formulation reduce overall complexity.' It is recommended to substantiate this claim with both theoretical analysis and quantitative experiments.
2. What are the main contributions or innovations of the embedding space division and dynamic equation-based approach compared to existing methods?
3. For different generation methods in Pseudo-cluster center constraints, could the authors provide theoretical analysis alongside the empirical results?
4. It is recommended to include more comparison methods from recent 2024 publications.
5. A discussion on the limitations of the proposed approach would be valuable.
6. Thorough proofreading and refinement of illustrations would enhance the clarity and quality of the paper.

**Questions:**

- What are the main contributions or innovations of the embedding space division and dynamic equation-based approach compared to existing methods?
- For different generation methods in Pseudo-cluster center constraints, could the authors provide theoretical analysis alongside the empirical results?

---

### Official Review · Reviewer_EBYD · 2024-11-02

**Soundness:** 3
**Presentation:** 2
**Contribution:** 3
**Rating:** 5
**Confidence:** 4

**Summary:**

The paper presents a novel implicit transferability modeling approach to address the challenges in estimating transferability of pre-trained models due to the varied architectures and training strategies. By incorporating newly proposed modeling and optimization strategies, the resultant framework demonstrates superior performance than existing methods across ten datasets with various model architectures and pre-training paradigms without the need of extensive training and computational resources.

**Strengths:**

- The proposed Implicit Transferability Modeling (ITM) framework introduces a fundamentally new approach to transferability estimation, distinguishing itself from existing models such as PED, SFDA, and LEAD. ITM not only improves model effectiveness but also addresses computational efficiency through the Divide-and-Conquer Adaptation (DCA) and Pseudo-Clustering Optimization (PCO) strategies, which optimize processing time and resource usage.
- The work further evaluates the applicability of transferability estimation on advanced architectures like Vision Transformers (ViT) and cutting-edge pre-training paradigms (e.g., MAE, SimMIM). This alignment with recent advancements in computer vision enhances the relevance and applicability of ITM within the current research landscape.
- The work is strengthened by rigorous mathematical derivations, which provide theoretical underpinnings that enhance clarity and make the methodological flow accessible and easy to follow.

**Weaknesses:**

- Figures are not well-used in the paper.
    - Too many figures (i.e., Figure 1 and 2) are used to describe the well-known problems in the field of transferability estimation.
    - Figure 1 is hard to understand at the first glance, particularly when it includes many abbreviations that are explained in the later literature.
    - Too much contents are included in the Figure 3, making it difficult to read
    at the first glance.
- There are several confusions in the paper:
    - If Equation (7) is re-written from Equation (6) by denoting C, then a $W^n$ term
    is missing in the formula. I doubt the mathematical correctness of Equation (7)
    and the related discussion on the benefits of this recursive form.
    - In Section 3.4, it has been mentioned that $\lambda$ is a hyper-parameter and
    is controlled by C. However, the paper doesn't specify how this is achieved.
    - In Section 3.4, it has been mentioned that the embedding pre-standardization
    is applied for various benefits. However, it doesn't specify how this is achieved
    as well.
    - In Section 3.4, it has been mentioned that the iteration number in DCA is fixed
    to 1. However, it doesn't specify the purpose of this setting. Also, if there is
    only one iteration required, what would be the benefits of transforming the update
    of the embedding space into a recursive formulation as in Equation (7)?
    - In the Conclusion, it has been claimed that "We conduct experiments on recent
    models and a diverse set of downstream tasks,...". However, experiments are only
    conducted on the image classification task.

**Questions:**

- I kindly suggest resolve the confusions mentioned in the Weakness part for the
overall clarity of the work.
- While Figure 7 is placed in the appendix, it can effectively offer insights into
the evolution of the embedding distribution. I kindly suggest to put it to the
most visible point in the paper and compare the evolution process of the ITM and
other existing transferability estimation approaches to demonstrate the
superiority of the proposed framework visually.
- It might be better to move the Table 4 into the main content to better support
the effectiveness of the proposed framework.
- Since LEAD is cited and can outperform the evaluated PED and SFDA, I am interested
in whether IMT could outperform it.
- What would happen if n is not fixed to 1? Would this lead to better results at
the cost of higher computational efficiency? I kindly suggest to carry out
experiments to discuss this in more detail, as this iterative optimization is the
core part of the DEA.
- The PCO strategy uses a strong inductive bias that the static distribution of
each class should be well-separated. Since this might not be always valid, I am
very interested in the applicability of IMT to other computer vision tasks like
image segmentation. Would there be any modifications to the framework to facilitate
applying it to other downstream tasks?

---

### Note · Authors · 2024-11-14

**Comment:**

We are retracting our paper despite having confidence in our work.   Upon further reflection, we have realized that the article is not yet well presented.   As a result, we have made the decision to withdraw the manuscript.

We would like to express our sincere gratitude to the reviewers for their detailed and valuable feedback.      Their comments have been extremely helpful in identifying areas for improvement.

We plan to revise the paper based on the reviewers' suggestions.      We appreciate the time and effort invested by the reviewers in evaluating our work.

**Withdrawal Confirmation:**

I have read and agree with the venue's withdrawal policy on behalf of myself and my co-authors.